# Understanding the risks for post-disaster infectious disease outbreaks: a systematic review protocol

Gina E C Charnley  ,[1,2] Ilan Kelman,[3,4,5] Katy Gaythorpe,[1,2] Kris Murray[1,2,6]

¹Faculty of Medicine, School of Public Health, Department of Infectious Disease Epidemiology, Imperial College London, London, UK
²MRC Centre for Global Infectious Disease Analysis, Imperial College London, London, UK
³Faculty of Mathematical and Physical Sciences, UCL Institute for Risk and Disaster Reduction, University College London, London, UK
⁴Faculty of Population Health Sciences, UCL Institute for Global Health, University College London, London, UK
⁵University of Agder, Kristiansand, Norway
⁶MRC Unit The Gambia, London School of Hygiene and Tropical Medicine, Fajara, The Gambia

**Correspondence to**
Gina E C Charnley;
g.charnley19@imperial.ac.uk

## ABSTRACT

**Introduction** Disasters have many forms, including those related to natural hazards and armed conflict. Human-induced global change, such as climate change, may alter hazard parameters of these disasters. These alterations can have serious consequences for vulnerable populations, which often experience post-disaster infectious disease outbreaks, leading to morbidity and mortality. The risks and drivers for these outbreaks and their ability to form cascades are somewhat contested. Despite evidence for post-disaster outbreaks, reviews quantifying them have been on short time scales, specific geographic areas or specific hazards. This review aims to fill this gap and gain a greater understanding of the risk factors involved in these contextual outbreaks on a global level.

**Methods and analysis** Using the Preferred Reporting Items for Systematic Review and Meta-Analysis Protocols 2015 checklist and Khan's methodological framework, a systematic search strategy will be created and carried out in August 2020. The strategy will search MEDLINE, Embase and GlobalHealth electronic databases and reference lists of selected literature will also be screened. Eligible studies will include any retrospective cross-sectional, case–control or cohort studies investigating an infectious disease outbreak in a local disaster affected population. Studies will not be excluded based on geographic area or publication date. Excluded papers will include non-English studies, reviews, single case studies and research discussing general risk factors, international refugee camps, public health, mental health and other non-communicable diseases, pathogen genetics or economics. Following selection, data will be extracted into a data charting form, that will be reviewed by other members of the team. The data will then be analysed both numerically and narratively.

**Ethics and dissemination** Only secondary data will be used and there will be no public or patient involvement; therefore, no ethical approval is needed. Our findings will aim to be disseminated through a peer-reviewed journal. The authors intend to use the results to inform future mathematical modelling studies.

## INTRODUCTION

Disasters have many forms, including those related to natural hazards (eg, earthquakes, volcanic eruptions, storm surges, floods, droughts, heatwaves, tsunamis) and armed conflict (eg, terrorism, civil war, genocide,

### Strengths and limitations of this study

► The lack of temporal or geographic limits is a major strength of this study, as it allows the authors and readers to gain a global understanding.
► Studies identified will be subject to a quality appraisal, and only certain publication methods will be selected, making the studies more comparable.
► A limitation of this study is excluding papers without an abstract or full text in English, as this creates a language bias.
► This review will not create a complete list of disaster-related disease outbreaks as it will only search peer-reviewed literature, which may have a publication bias.

political riots).[1 2] Disaster monitoring systems have shown alterations in frequency and intensity trends.[3 4] A possible contribution for this global change is human-induced climate change, through rising carbon dioxide emissions. This may alter some hazard parameters.[5 6] For example:

► Natural hazards—sea level rise and warming temperatures may alter hurricane frequency and intensity.[7 8]
► Armed conflict—increased temperatures may alter drought frequency, potentially influencing conflict escalation.[9 10]

The potential for these trends to alter the vulnerability of disaster-affected populations demonstrates the importance of understanding the complexities of how disasters impact people. The Sendai framework for disaster risk reduction[11] states that understanding disaster risk is the number one priority and risk management strategies should protect people and their property, health, livelihoods and productive assets while promoting and protecting all human rights. Understanding and managing risks posed by disasters is therefore paramount to reaching the sustainable development goals.[12] A way in which disasters impact human health is via infectious disease outbreaks, which have been extensively reported.[13–16]

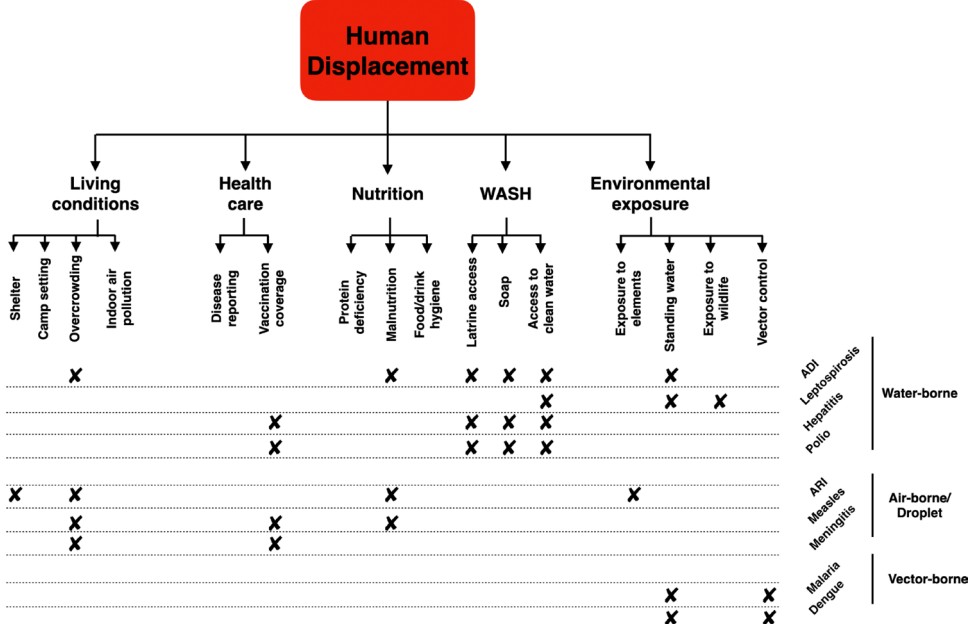

**Figure 1** Human population displacement as a possible risk factor cascade. ADI, acute diarrheal infection; ARI, acute respiratory infections; WASH, water, sanitation and hygiene (adapted from Kouadio *et al* and Hammer *et al*[25 27]).

Across diseases, outbreaks have a variety of factors capable of influencing initiation, duration and severity (eg, climate, food, water, sanitation, health systems) and can be exacerbated by complex interactions among such factors.[17] Understanding these causes is important for anticipating future disease risks in a rapidly changing world. After a disaster, the risk of infectious disease is low but real; for example, an outbreak of norovirus in Texas after Hurricane Katrina in 2005[18] and cutaneous leishmaniasis outbreaks beginning in 2013, during the Syrian conflict.[19] These outbreaks have a significant social and economic cost to already fractured communities and often show inequalities and insufficiencies that were present long before the outbreak or disaster. These issues show the complexity of outbreak risks in this context and the importance of identifying them.[20]

Several risk factors can lead to disease outbreaks following disasters, such as poor water, sanitation and hygiene conditions,[15] alterations in vector behaviour,[21] issues with housing and shelter,[22] problems obtaining healthcare[23] and mass population displacement.[16 24–26] To add further complexity, few risks act solely to cause an outbreak and risk factors are potentially linked, a concept known as risk factor cascades,[27] an example of which is shown in figure 1, using displacement as the cascade trigger. Examining these risk factors is very important in understanding population risk (risk=hazard×vulnerability×exposure). Several of these risks have the potential to increase vulnerability, and over-emphasis on a single hazard (such as a natural hazard or armed conflict) reduces this understanding.[28]

The intersection of disasters and disease outbreaks therefore provides a unique angle to understand the mechanisms through which global change can yield health impacts.[29 30] Alterations in disaster parameters mean that understanding the risks of disease outbreaks is crucial to preventing morbidity and mortality. Previous research that has attempted to collate individual disaster-related disease outbreaks has been on relatively short time scales (2000–2011),[25] specific geographic areas (Europe)[31] or focused on a certain hazard (tsunami).[32] This review aims to address this gap, to gain a global overview of these outbreaks and highlight potential areas where post-disaster relief needs to focus to reduce disease risk.

## AIMS AND OBJECTIVES

To identify disaster-related disease outbreaks and the risk factors that lead to these outbreaks via a comprehensive search of current literature. This will enable understanding of common etiological agents in disaster-related outbreaks, along with regions which are frequently impacted.

Objectives of this review are to:
1. Provide a global overview of infectious disease outbreaks that occurred in a post-disaster (natural hazard or armed conflict) setting, to examine common geographic regions, disasters and outbreak aetiologies.
2. Examine the risk factors that lead to these outbreaks and how they link to form cascades.
3. Suggest areas where global change, such as climate change, may exacerbate the identified risk factors.
4. Disseminate the review findings on global disaster-related disease outbreaks.

## METHODS

Systematic reviews are commonly used for evidence-based public health research, both for clinical practice

and biomedical research. They sit at the top of the hierarchy of evidence for medical research, as they are considered highly filtered research with low levels of bias.[33] For reviews to be classified as systematic they must first formulate questions, then appraise relevant studies on their quality and finally summarise the evidence found.[34] Indication in this case for a systematic review includes: (1) uncovering the international evidence, (2) identifying areas for further research and (3) investigating conflicting results.[35] An aetiology or risk systematic review most closely matches the review proposed here and is used to determine whether and to what degree a relationship exists between an exposure and a health outcome. For this review, the question needs to outline the exposure, disease and health condition of interest, the population and its location, and the study period where relevant. The types of studies included will mainly involve observational studies, namely retrospective, cross-sectional, case–control and cohort studies.[35 36]

## Protocol design
The Preferred Reporting Items for Systematic Review and Meta-Analysis Protocols 2015 checklist (see online supplemental appendix 1) will be used to develop the review protocol and will be guided by the methodological approach delineated by Khan *et al*.[37] The framework is set out to follow five stages: (1) framing the question(s), (2) identifying relevant work, (3) assessing study quality, (4) summarising the evidence and (5) interpreting the findings.

### Stage 1: framing the research questions
After preliminary research on natural hazards and armed conflicts and their risk factors for communicable disease outbreaks, it became apparent that quantification of these contextual outbreaks and their risks was insufficient to gain a clear global understanding of the issue. Due to this deficiency, the review questions are as follows (box 1):

### Stage 2: identifying relevant work
#### Search strategy
The following electronic databases will be searched: MEDLINE, Embase and Global Health, as these are appropriate for the subject matter, but no grey literature will be used. Reference lists of selected papers and reviews will also be screened for relevant papers and subject to the same screening process. Following the development of both key and Medical Subject Headings (MeSH) terms, the strategy and database choice will be reviewed by the Imperial College medical librarian, to ensure all relevant literature will be found. No standard definitions will be set

> ### Box 1 Research questions
> ▶ Which pathogens, disasters, global changes and geographic areas are commonly implicated in outbreaks in a post-disaster setting?
> ▶ Which risk factors are important in causing disaster-related disease outbreaks and how are they potentially linked to form cascades?

but instead terms will vary depending on the database and will be related to: (1) natural hazards, (2) armed conflict and (3) infectious disease outbreaks. Database-specific terms for natural hazards will be searched to including climatological, hydrological, geophysical, meteorological and armed conflict hazards, with a full search strategy shown in online supplemental appendix 2. No temporal or geographic limits will be set, and no specific risk factors will be searched to avoid bias in the search results. The systematic search will be carried out in August 2020 and additional searches may be requested later to identify any publications potentially missed. Results will be imported into Zotero reference management software.

Along with database-specific broad terms for outbreaks, specific epidemic potential diseases will also be searched, as identified by the WHO as common communicable disease outbreaks following disasters,[38] along with commonly reported diseases identified from preliminary scoping searches. Specific pathogens will include those capable of causing acute outbreaks but not causing an outbreak before the disaster. Therefore, despite evidence for contextual increases,[39 40] HIV, hepatitis B, hepatitis C and tuberculosis will not be searched, as they often cause more chronic disease and have a wide range of social implications beyond the scope of this study. Soft tissue, wound infections, inhaled fungal spores and aspiration pneumonia (tsunami lung) will also not be included, as such infections would only impact those that had open wounds and exposure to the pathogen in the environment. Therefore, they are not considered to have epidemic potential.

### Stage 3: assessing study quality
After the removal of duplicates using Zotero software, search results will be screened by one reviewer (GC). The aim will be to assess the study quality and decide on selection by comparing the publication against eligibility criteria (table 1). There are >100 critical appraisal tools published, and selecting the most useful is challenging; as there is no 'gold standard' for any study design or a widely accepted generic tool.[41] After consideration of published tools, the National Institute of Health (NIH) study quality assessment tools will be used. The NIH tool was chosen as it best captures the range of studies this research aims to review and accounts for bias and several methodological flaws.[42] A final score of either 'good' (>70% of questions answered 'yes'), 'fair' or 'poor' (<50% answered 'yes') will be given. All studies rating 'poor' will be removed and studies rated 'fair' will be assessed to try and decide if the questions answered 'no' would lead to major bias in answering the research question. One reviewer will be used based on time and personnel constraints of this study and it is acknowledged that this may increase the number of studies missed. Despite this, the authors still believe that the review is systematic as it aims to appraise and synthesis all available literature and provide sufficient detail to be reproducible. All titles and abstracts that meet the criteria will be subjected to full-text reading.

**Table 1** Eligibility criteria

| Inclusion | |
|---|---|
| Population | Any local population/community impacted by a disaster-related disease outbreak. |
| Intervention | Any investigation carried out to quantify a disease outbreak and understand the risk factors. |
| Comparator | Members of the disaster-affected population who did not acquire an infection during the outbreak. |
| Outcomes | ► The primary outcome is to understand disaster-related disease outbreaks on a global scale. <br> ► The secondary outcome consists of identifying the risk factors that result in these outbreaks. |
| Study type | ► Retrospective observational studies, namely, cross-sectional, case–control and cohort studies. <br> ► Full text or abstracts in English |
| **Exclusion** | |
| Papers without an explicit link between a disaster and an outbreak. | |
| Outbreaks in refugees/refugee camps, foreign armed forces, aid workers and international travellers, as this review aims to look at local outbreaks in regional populations. | |
| Non-English abstract and full texts, due to linguistic constraints. | |
| Review papers, as only primary sources are desired for this review. | |
| Single case reports in diseases not considered to have epidemic potential. | |
| Publications discussing general risk factors, public health, mental health and other non-communicable diseases, pathogen genetics or economic costs in a post-disaster setting, as these are beyond the scope of this review and its objectives. | |

A Preferred Reporting Items for Systematic Reviews and Meta-Analyses flow diagram will then be used, to show the number of publications selected and increase transparency.

It could be argued that if populations were displaced internationally by a disaster and an outbreak occurred, this may have been due to the disaster. Despite this, international refugee camps commonly housed refugees from multiple countries, impacted by multiple disasters; therefore, linking these outbreaks to specific disasters would be challenging. The only camp settings explored in this review will be national relief camps. By focusing on local populations, this will increase the likelihood that the outbreak was caused due to the disaster and not due to other external factors.

### Stage 4: summarising the evidence

A predetermined data charting form will be used based on preliminary reading and the objectives of the review. Extracted data will include information on the publication (title, authors, date, journal), disaster type, disease, case numbers, study area, study period, identified risk factors, methodological details (study design, sample sizes, laboratory tests, statistical analysis) along with any other relevant data. To ensure all relevant data are collected, the form will be reviewed by other members of the research team before implementation, and the data will be extracted independently by the reviewer (GC).

### Stage 5: interpreting the findings

Following the data extraction and to help illustrate how the information collected answers the aims and objectives, the results will be presented both: (1) numerically, with outbreaks broken down and quantified by geographic region, pathogen type and disaster type, along with statistical analysis to understand the importance of risk factors and (2) narratively, by synthesising the methods used, the importance of global change and the links between risk factors and possible cascades. As this is ongoing research, with no data currently collected, details of this stage remain in progress and will be finalised by August 2020. The ability to conduct a meta-analysis will depend on the volume and quality of literature that is returned from the searches and therefore will be decided after the data collection stage.

### Patient and public involvement

Patients and the public were not involved in this research, as the role of this study is to provide a protocol for a review using secondary data.

### ETHICS AND DISSEMINATION

Following review completion, a global overview of disaster-related disease outbreaks will be determined through time. To assess whether the research aims have been met, the authors will use the Grading of Recommendations, Assessment, Development and Evaluations framework.[43] If the aims are met, the study findings will allow the authors to determine important risk factors for these outbreaks and how they form cascades and links. It will also help form greater understanding of the observational methods used and how effective these have been in capturing and presenting these outbreaks. The results will determine important areas for future research by providing a greater understanding of how different geographic area and hazards impact disease outbreaks.

 Charnley GEC, *et al. BMJ Open* 2020;**10**:e039608. doi:10.1136/bmjopen-2020-039608

These findings will help to inform later quantitative studies into the dynamics of post-disaster outbreaks, by understand areas where further research is needed, and the best data is available.

As data collection for systematic reviews uses exclusively secondary data, no ethical approval is needed. The aim is to disseminate the review findings through publication in a peer-reviewed journal. Understanding how outbreaks impact vulnerable post-disaster populations and how under global changes this may be altered is very important in preventing mortality and morbidity. It is essential that the rights of these populations are protected and therefore their ability to protect their health in humanitarian crisis. We aim to use the knowledge from this review to guide future computational modelling studies into how displacement and its associated risk factors initiate and spread disease during post-disaster outbreaks.

**Acknowledgements** GECC, KAMG, KAM and IL acknowledge joint Centre funding from the UK Medical Research Council and Department for International Development [MR/R015600/1]. We would also like to thank the Imperial College medical librarian for his advice on the systematic search strategy.

**Contributors** GECC conducted preliminary research and searches, devised the search strategy and wrote the report, incorporated feedback and finalised the manuscript. KG revised the first draft and provided feedback based on her infectious disease expertise. IK revised the first draft and provided feedback based on his disaster and health expertise. KM provided supervision for the research activity planning, assisting in the conceptualisation of ideas and the overarching research goals and aims, along with providing feedback on the first draft.

**Funding** This work was supported by the National Environmental Research Council as part of the Grantham Institute for Climate Change and the Environment's (Imperial College London) Science and Solutions for a Changing Planet Doctoral Training Partnership [NE/S007415/1].

**Competing interests** None declared.

**Patient and public involvement** Patients and/or the public were not involved in the design, or conduct, or reporting, or dissemination plans of this research.

**Patient consent for publication** Not required.

**Provenance and peer review** Not commissioned; externally peer reviewed.

**ORCID iD**
Gina E C Charnley http://orcid.org/0000-0003-2087-7822

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
