## [Reviewer comments · BMJ Open]

ARTICLE DETAILS

TITLE (PROVISIONAL)	Understanding the risks for post-disaster infectious disease outbreaks: a systematic review protocol
AUTHORS	Charnley, Gina; Kelman, Ilan; Gaythorpe, Katy; Murray, Kris

VERSION 1 – REVIEW

REVIEWER	Abbas Ostadtaghizadeh Tehran University of Medical Sciences, IRAN
REVIEW RETURNED	18-May-2020

GENERAL COMMENTS	While the manuscript is a systematic review protocol no statistics, results and discussion in needed to include in the manuscript at this stage.
--

REVIEWER	Dell Saulnier Karolinska Institutet, Sweden
REVIEW RETURNED	12-Jun-2020

GENERAL COMMENTS	Review: bmjopen-2020-039608 Title: "Understanding the risks for post-disaster infectious disease outbreaks: a systematic review protocol" Dear authors, Thank you for an interesting and highly relevant protocol! Your review aims to understand risk factors for infectious disease outbreaks post-disaster. This is a substantial and well-known knowledge gap and conducting a high quality systematic review would be an excellent contribution. You propose a systematic review of English language articles on infectious diseases post-disaster and displacement at a global level from three databases, analyzed narratively and by meta-analysis, if possible (or "numerically"). I have a few suggestions to the methodology and rationale that I hope will improve the review's quality and the protocol's clarity. Major comments 1. You discuss displacement as an important factor for outbreaks throughout the abstract and introduction, but displacement is not present in the title, aim, or objectives. You instead list "risk factors" for outbreaks. Your research questions ask if displacement is important to outbreaks as well as what other risk factors exist. Am I correct to assume that you want to study displacement and its consequent risks (e.g. overcrowding) as the major risk factors for outbreaks? Or are you searching for any risk factors, regardless of their relationship to displacement? I think it would be useful before you begin searching and screening articles to clarify how you are conceiving displacement as a risk cascade (partially done in Figure 1, which I quite like) and make this clearer in the introduction, title, aim, and objective.2. What is "human-induced global change"? You liken it to climate change in the introduction (Page 3, Line 60 and Page 4, Lines 2-6)
--

and state later that you want to understand the mechanisms through which climate change can yield health impacts (Page 4, Lines 39-40). Is human-induced global change climate change then? If not, please clarify.

3. Please clarify your argument about how displacement risk factors relate to vulnerability, hazard, and exposure in the paragraph on displacement (Paragraph 3, Page 4).

4. Page 4, Line 43-44: You state that previous research has been on a short time scale. Can you please add to the methods the length of time between disaster and outcome that you expect to study, or the length of the short time scales that have already been studied?

5. One of your objectives is to “quantify which geographic areas and etiological agents commonly impact disaster-affected communities” (Page 4, lines 57-58). I wonder if you will be able to quantify regions and agents in a meaningful way, as you run the risk of instead quantifying the number of studies from a region or agent because of publication bias and the expected heterogeneity in study design, exposures, and outcomes. A word like ‘describe’ may be a more suitable choice.

6. You have selected three databases to search. Could you explain why you chose these three, and why you have not included Web of Science?

7. Why are the authors only including papers affecting local populations? If your assumption is that displacement is an important risk factor for outbreaks, why exclude refugee and internally displaced populations?

8. Although I understand the reason that the authors do not want to pre-define hazards and outbreaks, I believe they need to consider this for the sake of transparency and reproducibility during screening. How will the authors decide to include or exclude, for example, a study on a period of extreme heat that does not self-describe as a disaster, or diseases with epidemic potential, like cholera, where a single case constitutes an outbreak in most surveillance and reporting systems?

9. Articles are normally screened by two people for systematic reviews to improve the quality of the screening process. Can the authors explain the choice to use one screener?

10. The NIH quality assessment tools are a good choice, but how will the authors set the quality thresholds?

11. In your inclusion criteria, comparators will be affected persons who do not acquire an infection. Have the authors considered including non-affected populations and pre-/post- measures as comparators?

12. Related to comment #5, the authors should consider a narrative synthesis approach for analysis. It seems unlikely that there will be enough homogeneity to conduct a meta-analysis or other robust statistical analysis.

a. Popay et al. Guidance on the conduct of narrative synthesis in systematic reviews. 2006.

Minor comments

1. Page 5, Line 43: How will you choose which papers are selected for a reference list search?

	2. In the abstract, please briefly list your eligibility criteria rather than stating that inclusion/exclusion criteria will guide eligibility (Page 3, Line 30-33). 3. You could consider using the MOOSE reporting guidelines for your review rather than PRISMA. a. Stroup DF, Berlin JA, Morton SC, et al. Meta-analysis of observational studies in epidemiology: a proposal for reporting. Meta-analysis Of Observational Studies in Epidemiology (MOOSE) group. JAMA. 2000;283(15):2008-2012. doi:10.1001/jama.283.15.2008 4. Page 4, Line 16: What do you mean by “influences”? 5. Page 6, Line 3: Clarify in the rationale or earlier in the methods that you are searching only for disease that can cause a widespread outbreak within a population. 6. Consider combining the inclusion and exclusion criteria into a single table on eligibility criteria.
--	--

REVIEWER	Miguel Antonio Salazar Heidelberg Institute of Global Health, Germany
REVIEW RETURNED	05-Aug-2020

GENERAL COMMENTS	This research is needed in the nexus of the fields of infectious disease epidemiology, health emergencies and disaster risk management, and climate change and health. The study protocol is written clearly and describes the process of the systematic review well. The limitations on particular statistical analyses for interpretation of findings are explained to be based on researches found thus could not be expounded on.
---

VERSION 1 – AUTHOR RESPONSE

Reviewer Feedback Response - Understanding the risks of post-disaster infectious disease outbreaks: a systematic review protocol.

Reviewer 2

Major comments:

1. Displacement was suggested as an important risk factor in the preliminary reading, which is why it took a focus when writing the review protocol. I agree this may make the protocol confusing, as all risk factors want to be understood and explored in this review, to help understand how and if displacement is important, as suggested. The research aims, abstract and the introduction have therefore been altered but the title has remained the same, as risk factors as a whole will be explored, which I believe the title suggests.
2. All aspects of global change will be open to discussion in the final review. In the instance referred to, only climate change through rising carbon dioxide emissions was being discussed, this has now been made clearer.
3. The paragraph on mass displacement has been changed due to comment 1 and helps to alleviate issues raised in comment 3.
4. The time scale and geographic region of the previous studies have now been specified.

5. Aim 1 has been altered and a word on publication bias has been inserted into the “strength and limitations of this study” section, as this has been considered by the authors.

6. Database choice was reviewed and discussed with the Imperial College Medical librarian, as they have experience in conducting reviews with a biomedical focus. Databases were selected based on the research topic and it is believed that the three chosen, along with reference list searching would find all key papers.

7. A paragraph on why internationally displaced refugees and camps will be excluded is provided under the inclusion and exclusion table. The authors feel that this is a large topic and possibly beyond the scope of this review, perhaps requiring one in itself. Local populations were only looked at to avoid external factors influencing disease. For example, the relief workers which were implicated in the Haitian cholera outbreak did not contract the bacteria themselves due to the earthquake, despite the conditions created potentially exacerbating the transmission.

8. Terms for hazards and outbreaks were mainly set based on the MeSH terms of the database, this has been made more explicit now in the methods and refers the reader the appendices for those specific terms. Preliminary scoping searches suggested that single case studies were not common in diseases of epidemic potential and instead in particularly uncommon diseases. The exclusion criteria had been altered.

9. The use of two reviewers is both time and resource expensive and in this case, a second reviewer could not be found, a statement has been inserted stating this and acknowledging it as a limitation. It is understood that not having a second reviewer can increase the number of studies missed [1]. Despite this, the specificity of studies required for this review may reduce the need for a second reviewer, as this would reduce contention over which to include. Some authors suggest that to be ‘systematic’, a second reviewer is needed; therefore, the title of the protocol can be changed to literature review instead of systematic review if needed. Although, definitions of a systematic review do not state that a second reviewer is needed [2] (despite their common use) and the authors believe that this review will still be systematic as it aims to appraise and synthesise all available literature and will arguably meet the BMJ Best Practice for Appraising systematic reviews criteria [3].

10. More detail has been given on the use of the NIH assessment tool and the thresholds set.

11 & 12. Yes, non-affected and pre-disaster populations have been considered. As this would be helpful for a meta-analysis of the results. It was decided against, as this would be a large task for a review of this scope, and perhaps would be a future area of research on a subset of the review results. This also applies to the comments on meta-analysis, and it is expected that most meta-analysis will not be suitable, due to poor comparability and homogeneity between studies.

Minor comments:

3. PRISMA was used rather than MOOSE, as this is strongly encouraged in the BMJ Open author submission guidelines.

[1] Stoll, C.R., Izadi, S., Fowler, S., Green, P., Suls, J. and Colditz, G.A., 2019. The value of a second reviewer for study selection in systematic reviews. *Research Synthesis Methods*, 10(4), pp.539-545.

[2] Grant, M.J. and Booth, A., 2009. A typology of reviews: an analysis of 14 review types and associated methodologies. *Health Information & Libraries Journal*, 26(2), pp.91-108.

[3] BMJ Best Practice., 2020. *Appraising systematic reviews*. [On-line]. Available from: <https://bestpractice.bmj.com/info/toolkit/learn-ebm/appraising-systematic-reviews/> (accessed 11 Aug 2020).

VERSION 2 – REVIEW

REVIEWER	Dell Saunier Karolinska Institutet, Sweden
REVIEW RETURNED	17-Aug-2020
GENERAL COMMENTS	Dear authors, I have no further comments and recommend the protocol is published!